# Frequency- and spike-timing-dependent mitochondrial Ca²⁺ signaling regulates the metabolic rate and synaptic efficacy in cortical neurons

Ohad Stoler[1][†], Alexandra Stavsky[1][†], Yana Khrapunsky[1], Israel Melamed[2], Grace Stutzmann[3], Daniel Gitler[1], Israel Sekler[1]*, Ilya Fleidervish[1]*

[1]Department of Physiology and Cell Biology, Faculty of Health Sciences and Zlotowski Center for Neuroscience, Ben–Gurion University of the Negev, Beer Sheva, Israel; [2]Department of Neurosurgery, Faculty of Health Sciences and Zlotowski Center for Neuroscience, Ben–Gurion University of the Negev, Beer Sheva, Israel; [3]Rosalind Franklin University of Medicine and Science, Chicago Medical School, Center for Neurodegenerative Disease and Therapeutics, North Chicago, United States

*For correspondence:
sekler@bgu.ac.il (IS);
ilya@bgu.ac.il (IF)

# Equal contribution

Competing interest: The authors declare that no competing interests exist.

**Abstract** Mitochondrial activity is crucial for the plasticity of central synapses, but how the firing pattern of pre- and postsynaptic neurons affects the mitochondria remains elusive. We recorded changes in the fluorescence of cytosolic and mitochondrial Ca²⁺ indicators in cell bodies, axons, and dendrites of cortical pyramidal neurons in mouse brain slices while evoking pre- and postsynaptic spikes. Postsynaptic spike firing elicited fast mitochondrial Ca²⁺ responses that were about threefold larger in the somas and apical dendrites than in basal dendrites and axons. The amplitude of these responses and metabolic activity were extremely sensitive to the firing frequency. Furthermore, while an EPSP alone caused no detectable Ca²⁺ elevation in the dendritic mitochondria, the coincidence of EPSP with a backpropagating spike produced prominent, highly localized mitochondrial Ca²⁺ hotspots. Our results indicate that mitochondria decode the spike firing frequency and the Hebbian temporal coincidences into the Ca²⁺ signals, which are further translated into the metabolic output and most probably lead to long-term changes in synaptic efficacy.

## Introduction

For the neuronal circuit to function properly, the energy demand in all compartments of the individual neurons needs to be precisely matched by local, primarily mitochondrial (*Celsi et al., 2009*; *Rangaraju et al., 2019*), ATP production. During periods of enhanced neuronal activity, mitochondria accelerate ATP production by allowing the cytosolic Ca²⁺ elevations to propagate into the mitochondrial matrix (*Ashrafi et al., 2020*; *Díaz-García et al., 2021*). When cytosolic $[Ca^{2+}]_i$ rises, Ca²⁺ ions, powered by the steep mitochondrial membrane potential, flow into the mitochondrial matrix via the Ca²⁺ uniporter MCU (*Baughman et al., 2011*; *De Stefani et al., 2011*), and are then extruded back into the cytosol by the mitochondrial Na⁺/Ca²⁺ antiporter NCLX (*Palty et al., 2010*). The mitochondrial Ca²⁺ elevation increases ATP production by activating at least three Krebs cycle enzymes (*De Stefani et al., 2016*; *Wescott et al., 2019*). When Ca²⁺ signaling is disrupted, mitochondria in presynaptic terminals fail to maintain the stable ATP concentration during enhanced activity periods (*Ashrafi et al., 2020*; *Giorgio et al., 2013*; *Glancy and Balaban, 2012*). The link between the distinctive types of the neuronal

electrical activity, mitochondrial $Ca^{2+}$ signaling and metabolism in other neuronal compartments is poorly understood, however.

Recent evidence using photo-uncaging of glutamate on spines of cultured hippocampal neurons indicates that dendritic mitochondria play a critical role in long-term regulation of synaptic strength (*Rangaraju et al., 2019*). Thus, local inhibition of the mitochondria by a phototoxic protein abolishes the synaptic plasticity in the affected dendritic segment. Under physiological conditions, the plastic changes in synaptic efficacy are believed to be controlled by the relative timing of pre- and postsynaptic action potentials (APs) (*Bi and Poo, 1998*; *Holtmaat and Svoboda, 2009*; *Markram et al., 1997b*). Spike-timing–dependent plasticity (STDP) relies on activation of the postsynaptic NMDA receptors, which are $Ca^{2+}$ permeable and require both glutamate and depolarization to open (*Nowak et al., 1984*). Mitochondrial handling of $Ca^{2+}$ ions is known to have a significant effect on cytosolic $Ca^{2+}$ dynamics (*Rizzuto et al., 2004*; *Szabadkai and Duchen, 2008*). The mitochondrial depletion, however, had no effect on glutamate evoked cytosolic $Ca^{2+}$ dynamics (*Rangaraju et al., 2019*), indicating that these organelles play a yet unexplored downstream role in the cascade of reactions leading to STDP.

Here, using whole-cell electrical recordings from Layer 5 pyramidal neurons in cortical slices and fluorescence imaging of cytosolic and mitochondrial $Ca^{2+}$ indicators, we show that single or few spikes trigger rapidly rising and decaying mitochondrial $Ca^{2+}$ elevations in all neuronal compartments, with kinetics similar to cytosolic $Ca^{2+}$ transients. Our evidence indicates that the mitochondria's $Ca^{2+}$ signaling and metabolic rate depend critically on spike firing frequency. We further report that, in dendrites, the coincidence of unitary EPSP and a backpropagating action potential produces a localized $[Ca^{2+}]_m$ transient, which, in addition to enhancing local ATP synthesis, could play a role in STDP.

## Results

### Spike-elicited mitochondrial $Ca^{2+}$ transients

To determine how neuronal electrical activity affects mitochondrial $Ca^{2+}$ dynamics, we performed somatic whole-cell recordings from L5 pyramidal neurons expressing the mitochondria-targeted, neuron-specific $Ca^{2+}$ indicator, mitoGCaMP6m. The intracellular solution was supplemented with the cytosolic $Ca^{2+}$ indicator, Fura-2, that could be excited separately from mitoGCaMP6m. Series of thin, high-resolution optical sections over the vertical extent of the neuron, representing Fura-2 and mitoGCaMP6m fluorescence elicited by two-photon excitation at 760 and 960 nm, respectively, were used to reconstruct the morphology of the neuron and to reveal the position of the individual mitochondria within its soma and processes. To improve the temporal resolution of the optical signals, dynamic fluorescence measurements were obtained from the smaller regions of interest in soma, axon, and dendrites. In a typical experiment, two APs elicited by the injection of two brief current pulses via the patch pipette caused cytosolic and mitochondrial $Ca^{2+}$ elevations in the soma of an L5 cell (*Figure 1a*). Although the cytosolic $Ca^{2+}$ signals had relatively even intensity, the change in mitoGCaMP6m fluorescence occurred at 'hotspots', each representing an individual mitochondrion. The mitochondrial $Ca^{2+}$ elevations began with a short delay after the beginning of the spike train, and they were observed only in the electrically active neurons. At the same time, the fluorescence of the nearby mitoGCaMP6m expressing mitochondria belonging to the non-active cells did not change (*Figure 1—figure supplement 1*).

The AP-elicited mitochondrial $Ca^{2+}$ elevations required $Ca^{2+}$ influx from the extracellular space. It is likely that $Ca^{2+}$ ions enter via the voltage-gated $Ca^{2+}$ channels, as bath application of $Cd^{2+}$ (200 μM) inhibited both cytosolic and mitochondrial $Ca^{2+}$ transients (*Figure 1—figure supplement 2*). Previous studies have shown that $Ca^{2+}$ depletion of the endoplasmic reticulum (ER) in cultured neurons does not influence mitochondrial $Ca^{2+}$ elevations (*Ashrafi et al., 2020*). Consistent with these results, we found that blockade of ER $Ca^{2+}$ channels by Ryanodine (100 μM) and Dantrolene (100 μM) (*Figure 1—figure supplement 3*) had no significant effect on mitochondrial $Ca^{2+}$ transients.

An increase in the number of spikes produced progressively larger mitoGCaMP6m transients in the soma (*Figure 1b*) and the apical dendrite (*Figure 1—figure supplement 4*) with a progressively slower rising phase. Thus, the mito-$Ca^{2+}$ signals elicited by 50 APs grew throughout the spike train duration, reaching a peak ΔF/F value of 59% ± 11% (n = 29) and 54% ± 11% (n = 22) for soma and apical dendrite, respectively, at 120 ± 18ms (n = 10) after the train end. In contrast with previous reports (*Ashrafi et al., 2020*; *Lin et al., 2019*), mito-$Ca^{2+}$ transients elicited by 2–20 spikes decayed

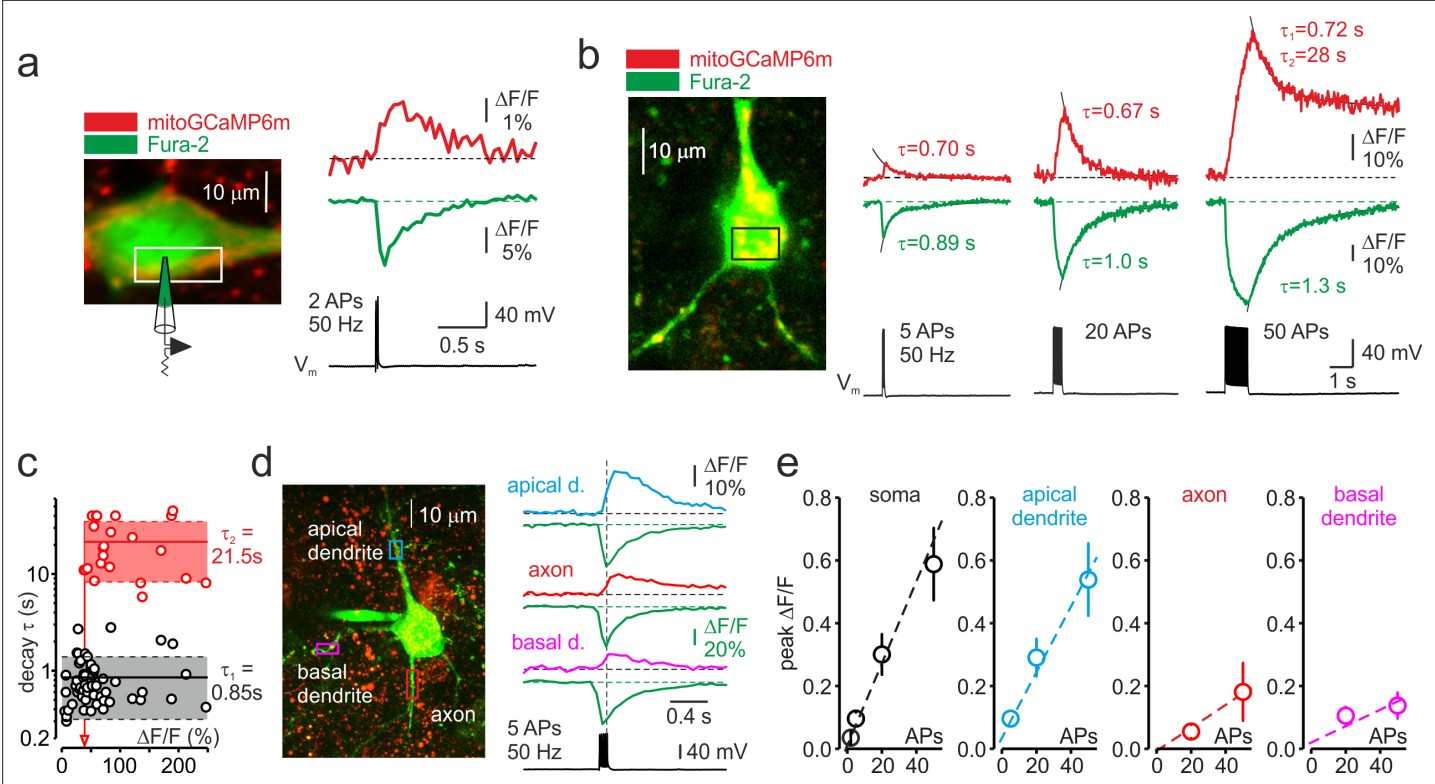

**Figure 1.** Trains of action potentials elicit mitochondrial Ca²⁺ elevations in soma and processes of L5 pyramidal neurons. (**a**) Changes in mitoGCaMP6m and Fura-2 fluorescence elicited by a train of two APs in a representative L5 pyramidal cell. *Left,* The image obtained by merging two optical sections through part of a L5 neuron at excitation wavelengths of 760 and 960 nm, eliciting the Fura-2 and mitoGCaMP6m fluorescence, respectively. *Right,* Somatic mitoGCaMP6m (red) and Fura-2 (green) ΔF/F transients elicited by a train of two APs. MitoGCaMP6m trace (red) is an ensemble average of 16 consecutive sweeps. (**b**) Small-amplitude mitoGCaMP6m transients decay rapidly. *Left,* Localization of the mitoGCaMP6m labeled mitochondria in a representative L5 neuron. The image was obtained by merging the fourteen, 1 μm thick optical sections at excitation wavelengths of 760 and 960 nm, eliciting the Fura-2 and mitoGCaMP6m fluorescence, respectively. The yellow color indicates colocalization of the Fura-2 (green) and mitoGCaMP6m (red) fluorescence, revealing the position of the mitochondria. The rectangle indicates the region of interest from which the optical sweeps were obtained. *Right,* somatic mitoGCaMP6m (red) and Fura-2 (green) transients elicited by 5, 20, and 50 APs in a cell shown on the left. Black lines are the best single or double exponential fits of the decay. Notice that after the relatively small [Ca²⁺]ₘ elevations evoked by five or 20 APs, the Ca²⁺ clearance is rapid and follows the single exponential time course (τ ~ 0.7 s). The decay of larger transients evoked by 50 APs is biexponential ($\tau_1$ of ~0.7 s and $\tau_2$ of ~28 s). (**c**) Decay time course of mitochondrial Ca²⁺ transients depends on their amplitude. Each dot represents a decay time constant of the mitoGCaMP6m transient ($\tau_1$ for monoexponential, $\tau_1$ and $\tau_2$ for bi-exponential decays) obtained in recordings from 42 neurons and plotted against its peak amplitude. Continuous lines are mean $\tau_1$ (n = 55, black) and $\tau_2$ (n = 23, red); dashed lines represent standard deviation from the mean. Arrow represents the amplitude of the smallest mitoGCaMP6m transient (ΔF/*F* = 44 %), which decayed biexponentially. (**d**) The amplitude of spike-evoked mitoGCaMP6m fluorescence transients varies between different neuronal processes. *Left,* Localization of the mitoGCaMP6m labeled mitochondria in a representative L5 pyramidal neuron. The rectangles indicate the regions within the apical dendrite (cyan), basal dendrite (magenta), and axon initial segment (red) from which fluorescence measurements were obtained. *Right,* Mitochondrial and cytosolic (green) Ca²⁺ transients elicited by a train of 5 APs at 50 Hz in different neuronal compartments. (**e**) Mean peak ΔF/F of the mitoGCaMP6m transients as a function of the number of action potentials in soma, apical, and basal dendrites, and axon initial segment. Shown are mean values ± SE (n = 5–29).

The online version of this article includes the following figure supplement(s) for figure 1:

**Figure supplement 1.** Ca²⁺ elevations in the mitochondria of the electrically active neurons.

**Figure supplement 2.** Blockade of voltage-gated Ca²⁺ channels abolishes spike-evoked mitochondrial Ca²⁺ transients.

**Figure supplement 3.** Blockade of the ER Ryanodine receptors has no significant effect on mitochondrial Ca²⁺ transients.

**Figure supplement 4.** The amplitude of mitoGCaMP6m transients varies as a function of the number of APs.

**Figure supplement 5.** The decay time course of mitochondrial Ca²⁺ transients varies as a function of their amplitude.

**Figure supplement 6.** MitoGCaMP6m is homogenously expressed in soma and processes of L5 pyramidal neurons.

as rapidly or even faster than cytosolic $Ca^{2+}$ transients. However, following large elevations, $[Ca^{2+}]_m$ remained high for tens of seconds, reaching the resting level long after the cytosolic $Ca^{2+}$ concentration completely recovered. We systematically examined the relationship between the peak amplitude and the decay rate of the mito-$Ca^{2+}$ transients in 42 pyramidal neurons (*Figure 1c*, *Figure 1—figure supplement 5*). Decay of smaller transients (peak $\Delta F/F < 40\%$) was always monoexponential, with $\tau$ = 0.86 ± 0.11 s (n = 25) as well as decay of some middle-sized ($\Delta F/F$ 40–75%) transients ($\tau$ = 0.82 ± 0.10 s, n = 10). The decay of other middle size and large ($\Delta F/F > 75\%$) transients was double exponential, with $\tau_1$ = 0.86 ± 0.13 s and $\tau_2$ = 22.0 ± 2.85 s (n = 23). The decay of cytosolic $Ca^{2+}$ transients always followed a single exponential time course which was characterized by $\tau$ = 1.03 ± 0.14 s (n = 22) for small and $\tau$ = 1.19 ± 0.4 s (n = 20, p = 0.43) for large transients.

To elucidate the differences in mitochondrial $Ca^{2+}$ signaling between distinct neuronal compartments, we monitored mitoGCaMP6m fluorescence in ~10 µm long regions of interest in basal, apical dendrites, and axon initial segments (AISs) during trains of five APs (*Figure 1d*). While the amplitude and time course of the cytosolic $Ca^{2+}$ responses in all these compartments were similar, the magnitude of mitochondrial signals was remarkably polar. The amplitude of the mitochondrial $Ca^{2+}$ elevations in apical dendrite was as prominent as in the soma. In contrast, in the AIS and thin basal dendrites, the mito-$Ca^{2+}$ responses were dramatically smaller.

We next sought to evaluate the relationship between the number of APs and the mean peak amplitude of the mitoGCaMP6m transients in the soma, AIS, apical and basal dendrites of 29 neurons (*Figure 1e*). The steepness of this relationship was greater in soma and apical dendrites (1.3% and 1.1% $\Delta F/F$ per spike, respectively) compared with AIS and basal dendrites (0.4% and 0.3% $\Delta F/F$ per spike, respectively). The compartmental differences in mitochondrial signals were not due to a different magnitude of cytosolic $Ca^{2+}$ elevations. The pattern of $[Ca^{2+}]_i$ during the neuronal activity is known to be complex. However, the differences in peak cytosolic $Ca^{2+}$ levels were subtle, and they poorly correlated with the magnitude of mitochondrial $Ca^{2+}$ elevations. For example, the peak $\Delta F/F$ amplitude of Fura-2 transients elicited by twenty spikes was 26% ± 3.6% (n = 22) for soma, 37% ± 4% (n = 13) for apical dendrite, 38% ± 4% (n = 13) for basal dendrites, and 25% ± 4% in the AIS (n = 7). The compartmental differences in magnitude of the mitoGCaMP6m transients could, at least partially, be explained by the lower expression level of fluorescence probe in the mitochondria localized within the thinner neuronal processes. This seems to be unlikely, however, since, in all neuronal compartments, maximal mitoGCaMP6m fluorescence of the individual mitochondria following prolonged depolarization of the neuronal membrane was similar (*Figure 1—figure supplement 6*). Our evidence, therefore, points to the existence of an as-yet-unidentified mechanism that differentially regulates the mitochondrial $Ca^{2+}$ entry in distinct neuronal compartments.

## Frequency-dependent amplification of mitochondrial $Ca^{2+}$ elevations

Next, we examined whether mitochondrial $Ca^{2+}$ elevations are sensitive to firing frequency. *Figure 2a* shows an optical recording from the soma of a representative L5 neuron in which cytosolic and mitochondrial $Ca^{2+}$ transients were elicited by trains of twenty APs at 20, 50, and 100 Hz. As spike frequency increased, the rise of cytosolic $Ca^{2+}$ transients became progressively faster and their peak amplitude modestly increased (*Figure 2—figure supplement 1*). In contrast, the mitochondrial $Ca^{2+}$ signals showed a very different frequency dependence. Firing at a frequency of 20 Hz or lower elicited a minimal elevation in $[Ca^{2+}]_m$, whereas the response to spikes at a frequency of 50 Hz or higher was dramatically greater. The steep frequency dependence of the mitochondrial signals was observed in all neuronal compartments, including somas, apical, and basal dendrites, making it unlikely that it reflects the frequency-dependent failure of AP backpropagation (*Spruston et al., 1995*). The frequency-dependent amplification of mitochondrial $Ca^{2+}$ transients was observed in all 21 neurons tested with either 50, 20, or 5 APs (*Figure 2b*). The mean ratio of peak amplitudes of the mitochondrial $Ca^{2+}$ transients elicited by trains of spikes at 50 and 20 Hz was larger when neurons were subjected to longer (3.36 ± 0.46 times, n = 22 for 50 APs) than to shorter (1.74 ± 0.18 times, n = 5 for 5 APs) trains (*Figure 2c*). Systematic varying of AP frequency in a range from 10 to 100 Hz revealed that the peak amplitude of mito-$Ca^{2+}$ transients behaved as Bolzmannian function of the frequency, with mean half-amplitude of 38 Hz and steepness of 22 $Hz^{-1}$ (n = 12).

The decay time course of the mitochondrial $Ca^{2+}$ transients elicited by high-frequency spike trains was significantly slower than those produced by the same number of spikes at a lower frequency

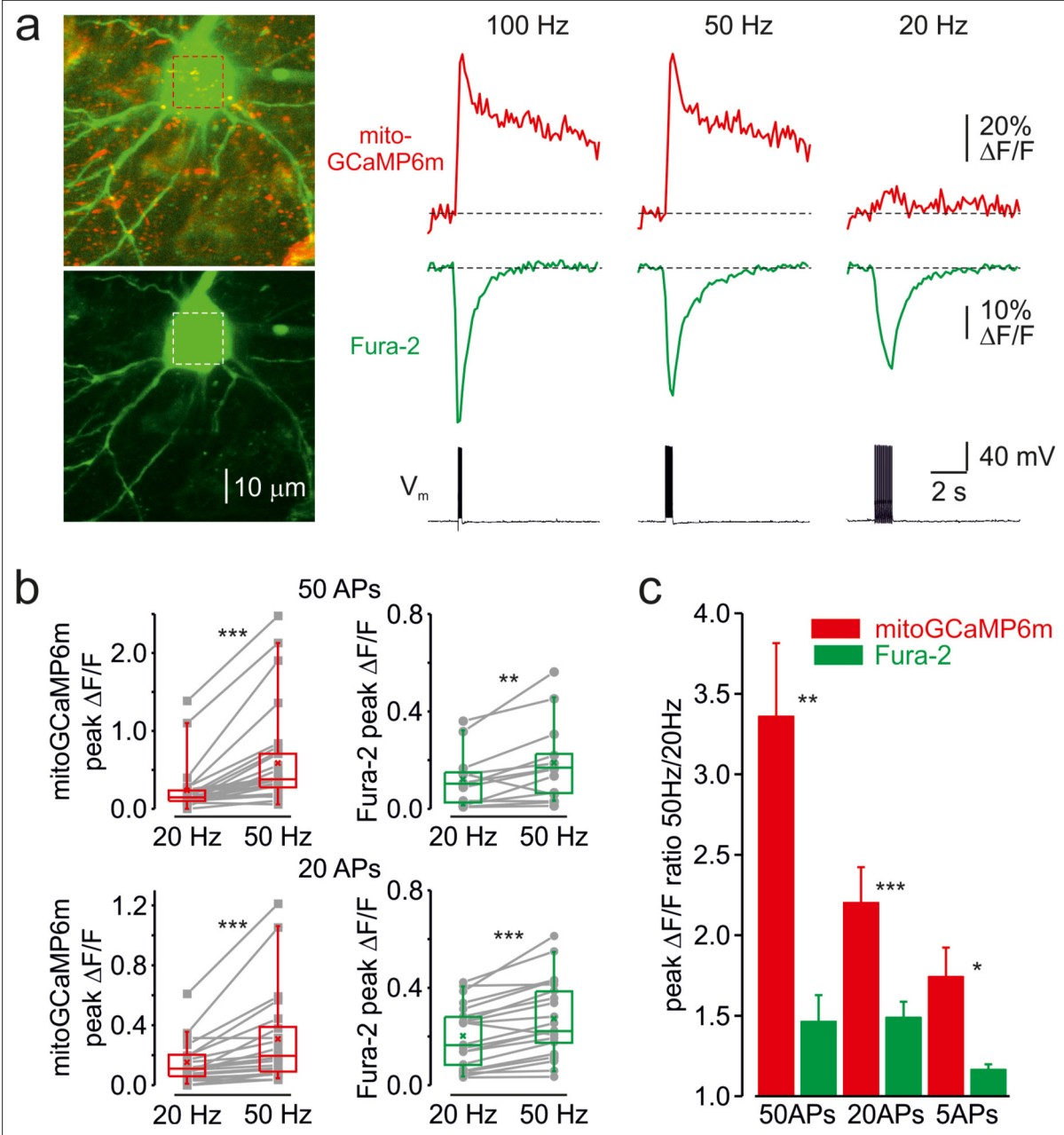

**Figure 2.** Frequency-dependent amplification of the mitochondrial Ca²⁺ elevations. (**a**) Cytosolic and mitochondrial Ca²⁺ elevations elicited at the soma of a representative pyramidal neuron by trains of 20 APs at 100, 50, and 20 Hz. *Left*, The top image is mitoGCaMP6m labeled mitochondrial map of a representative L5 pyramidal neuron, obtained by merging the Fura-2 and mitoGCaMP6m fluorescence (for detail, see ***Figure 1***). The bottom image is the maximum intensity Z-projection representing the Fura-2 fluorescence only. *Right*, The mitoGCaMP6m (red) and Fura-2 (green) ΔF/F transients elicited at the soma by the spike trains at indicated frequency. Notice that, at 20 Hz, the train of spikes produced almost no mitochondrial Ca²⁺ elevation, while the amplitude of the cytosolic Ca²⁺ transient changed little as a function of spike frequency. (**b**) Peak amplitude of mitoGCaMP6m (red) and Fura-2 (green) ΔF/F transients elicited by a train of 50 (top) or 20 (bottom) APs at a frequency of 20 and 50 Hz. The gray lines connect the paired values obtained from the same individual neuron at two firing frequencies. Box plots represent the 25–75% interquartile range, and the whiskers expand to the 5–95% range. A horizontal line inside the box represents the median of the distribution, and the mean is represented by a cross symbol (X). (**c**) Mean ratio of peak amplitudes of mitoGCaMP6m (red) and Fura-2 (green) ΔF/F transients elicited by trains of 50, 20, and 5 APs at 50 and 20 Hz frequency.

The online version of this article includes the following figure supplement(s) for figure 2:

**Figure supplement 1.** The amplitude of the mitochondrial Ca²⁺ elevation is proportional to the rate-of-rise of the cytosolic Ca²⁺ concentration.

**Figure supplement 2.** High-frequency firing elicits slowly decaying mitochondrial Ca²⁺ transients in soma and dendrites of L5 pyramidal neurons.

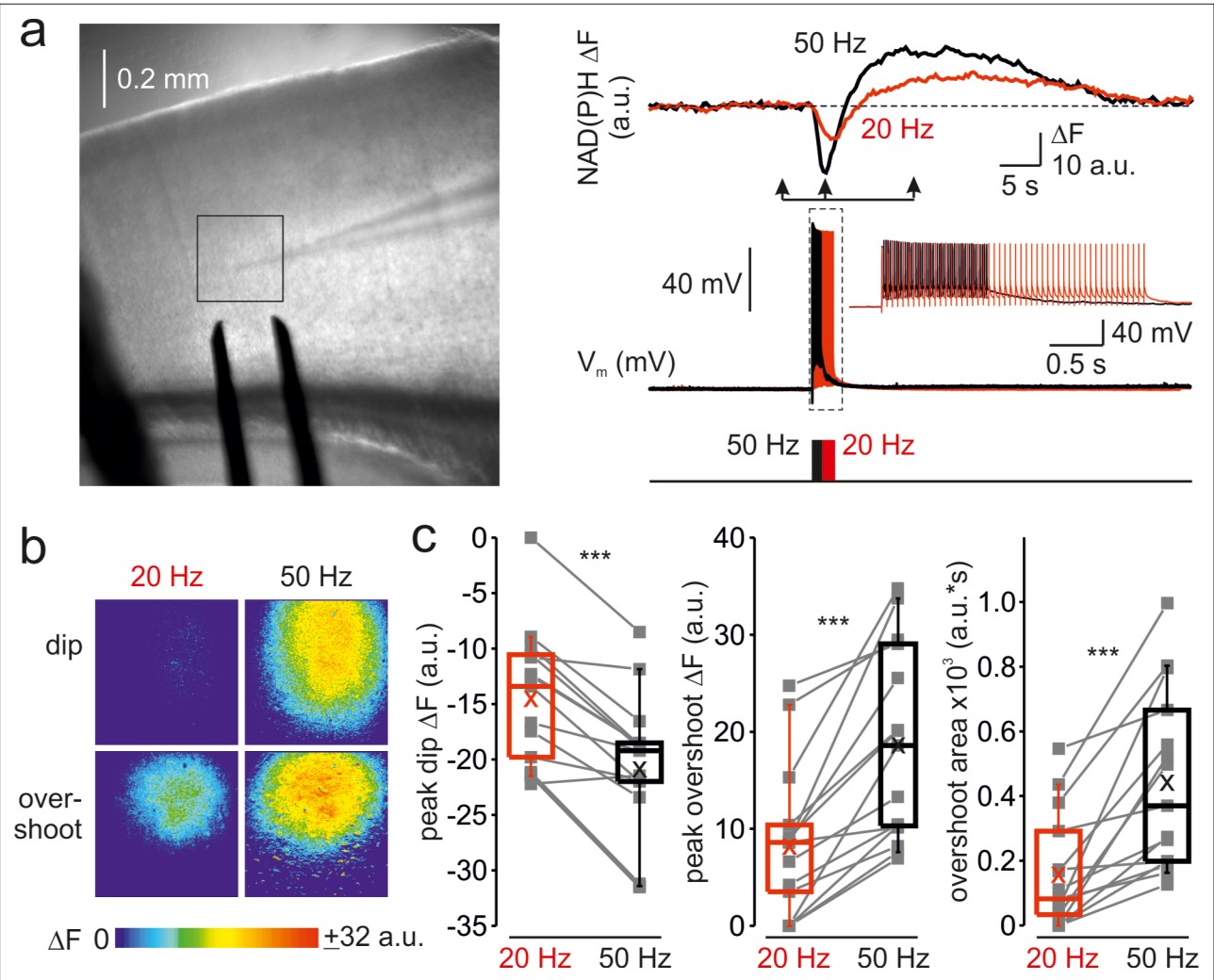

**Figure 3.** Frequency-dependent amplification of spike elicited changes in mitochondrial NAD(P)H auto-fluorescence. (**a**) In a representative cortical slice, changes in NAD(P)H fluorescence in response to extracellular stimuli trains depend on stimulation frequency. *Left,* DIC image of a coronal slice during the electrical and optical recording. The rectangle indicates the region from which the auto-fluorescence measurements were obtained. The stimuli were delivered via the bipolar electrode placed on the white-gray matter border, and the whole-cell recording was obtained from an L5 neuron within the same cortical column. *Right,* The membrane potential and optical traces evoked by trains of 50 just suprathreshold stimuli at 50 Hz (black) and 20 Hz (red). Notice that both dip and overshoot of the NAD(P)H signals are more prominent at 50 Hz. *Inset*: Stimuli intensity was carefully adjusted to elicit only a single AP per stimulus. (**b**) The amplitude of NAD(P)H signals depends on stimulation frequency, whereas their spatial extent does not. Shown are pseudocolor maps of change in the NAD(P)H fluorescence between the times marked by the arrowheads in **a**. (**c**) Higher frequency stimulation causes an increase in the magnitude of the dip and of the overshoot of the NAD(P)H signal. Box plots representing the peaks of the dip (left), the peaks of the overshoot (middle), and the area of the overshoot (right) of the NAD(P)H signals evoked by trains of 20 Hz (red) and 50 Hz (black) stimuli. The gray lines connect the paired values obtained from the same cortical regions at two firing frequencies (n = 15 ROIs, ten cortical slices, three mice). Box plots represent the 25–75% interquartile range, and the whiskers expand to the 5–95% range. A horizontal line inside the box represents the median of the distribution, and the mean is represented by a cross symbol (X).

The online version of this article includes the following figure supplement(s) for figure 3:

**Figure supplement 1.** Firing frequency dependent amplification of NAD(P)H signal in the CA1 area of hippocampus.

(*Figure 2—figure supplement 2*). Thus, trains of fifty APs at 50 Hz elicited the extremely slowly decaying mito-$Ca^{2+}$ transients ($\tau > 10$ s) in 11/15 somas and 7/10 apical dendrites. In contrast, all but one mito-$Ca^{2+}$ transient produced in the same neurons by fifty APs at 20 Hz decayed rapidly.

## Frequency-dependent acceleration of the mitochondrial NAD(P)H metabolism

To determine whether the frequency-dependent amplification of $Ca^{2+}$ signals in the mitochondria affects their metabolic activity, we monitored the changes in NAD(P)H autofluorescence elicited by trains of brief, just suprathreshold antidromic stimuli at a different frequency (*Figure 3a*). Whole-cell recording from a single pyramidal neuron within the region of interest was obtained to tune the stimuli intensity such that each stimulus would elicit only one AP. In cortical neurons, NAD(P)H signals primarily reflect changes in mitochondrial NAD(P)H pool (*Díaz-García et al., 2021*). In response to electrical stimulation, we observed a negative deflection in the NAD(P)H autofluorescence ('dip') which indicates an increased rate of electron transfer reflected in NAD(P)H oxidation, followed by a positive transient ('overshoot') which indicates Krebs-cycle-dependent replenishment of NAD(P)H pool. At higher stimulation frequency, the magnitude of both dip and overshoot of the NAD(P)H signals were enhanced, consistent with the previous reports that the rates of NAD(P)H oxidation and synthesis are dependent on the $Ca^{2+}$ level in the mitochondrial matrix (*Díaz-García et al., 2021*). Spatio-temporal analysis of the NAD(P)H autofluorescence dynamics at two stimulation frequencies (*Figure 3b*) revealed that the changes in the fluorescence were spatially restricted to Layer 5 of a single cortical column and that the frequency-dependent amplification of both dip and overshoot amplitude was relatively uniform within the stimulated region. A comparison of NAD(P)H autofluorescence changes in response to a train of 50 stimuli at 20 Hz and 50 Hz in ten cortical slices (*Figure 3c*) revealed the same frequency dependence as with $[Ca^{2+}]_m$. The dip amplitude increased from $-15 \pm 2$ a.u. at 20 Hz to $-21 \pm 2$ a.u. at 50 Hz (n = 15 ROIs, $p < 0.001$), the overshoot's peak amplitude increased from $8 \pm 2$ a.u. to $19 \pm 3$ a.u. ($p < 0.001$) and the overshoot area increased from $156 \pm 46$ a.u.·s to $441 \pm 71$ a.u.·s ($p < 0.001$), respectively. Because the glial responses might partially contaminate NAD(P)H signals obtained in the cortical slices, we tested the frequency dependence of NAD(P)H responses in the in Stratum pyramidale of CA1 area of the hippocampus (*Figure 3—figure supplement 1*) which predominantly contains neuronal cell bodies. As in the neocortex, NAD(P)H signals in response to trains of stimuli delivered to the Stratum oriens were significantly enhanced at the higher stimulation frequency.

## Localized dendritic $[Ca^{2+}]_m$ elevations elicited by the coincidence of postsynaptic AP and EPSP

We next sought to elucidate how synaptic activity affects mitochondrial $Ca^{2+}$ dynamics. After filling the cell for ~20 min to allow the diffusion of Fura-2 into the dendrites, we positioned the bipolar electrode close to an apical or basal dendrite. Delivery of a single brief stimulus (0.1ms), with its amplitude adjusted to keep the subsequent EPSP below the threshold for postsynaptic spike generation produced no detectable cytosolic or mitochondrial $Ca^{2+}$ signals (n = 7). The single AP elicited by brief somatic current pulse injection, however, produced a cytosolic but no mitochondrial $Ca^{2+}$ response in the dendrites. Remarkably, the coincidence of the EPSP and backpropagating AP synergized to elicit a robust cytosolic and mitochondrial $Ca^{2+}$ response (*Figure 4a*). While most dendritic mitochondria were silent, the EPSP and AP coincidence created single localized mitochondrial $Ca^{2+}$ "hotspots" with peak $\Delta F/F$ amplitude of $6.4\% \pm 0.8\%$ (n = 8) in the dendritic regions of interest. We interpreted the appearance of these hotspots as evidence for a highly restricted, probably single mitochondrial $Ca^{2+}$ elevation, in the vicinity of the active spine. The unitary character of the mitochondrial signals under this experimental paradigm can be explained by a spatial sparseness of the synapses formed by the presynaptic fiber on the dendrites of the postsynaptic cortical cell (*Markram et al., 1997a*) such that only one out of a few currently active spines could be found in the relatively short segment of a dendritic branch that we examined. The cytosolic $Ca^{2+}$ elevation, measured in the same hotspot at which the mitoGCaMP6m signal was detected, was not significantly larger than in the nearby dendrite. The failure to see the cytosolic $Ca^{2+}$ "hotspots" is, most probably, due to temporal and amplitude resolution of our optical recording that was insufficient to reveal $Ca^{2+}$ elevation in the tiny volume between the spine neck and the mitochondrion during the fast single spine $Ca^{2+}$ transient (*Miyazaki and Ross, 2017*; *Svoboda et al., 1996*). We extended our analysis by measuring the cytosolic and

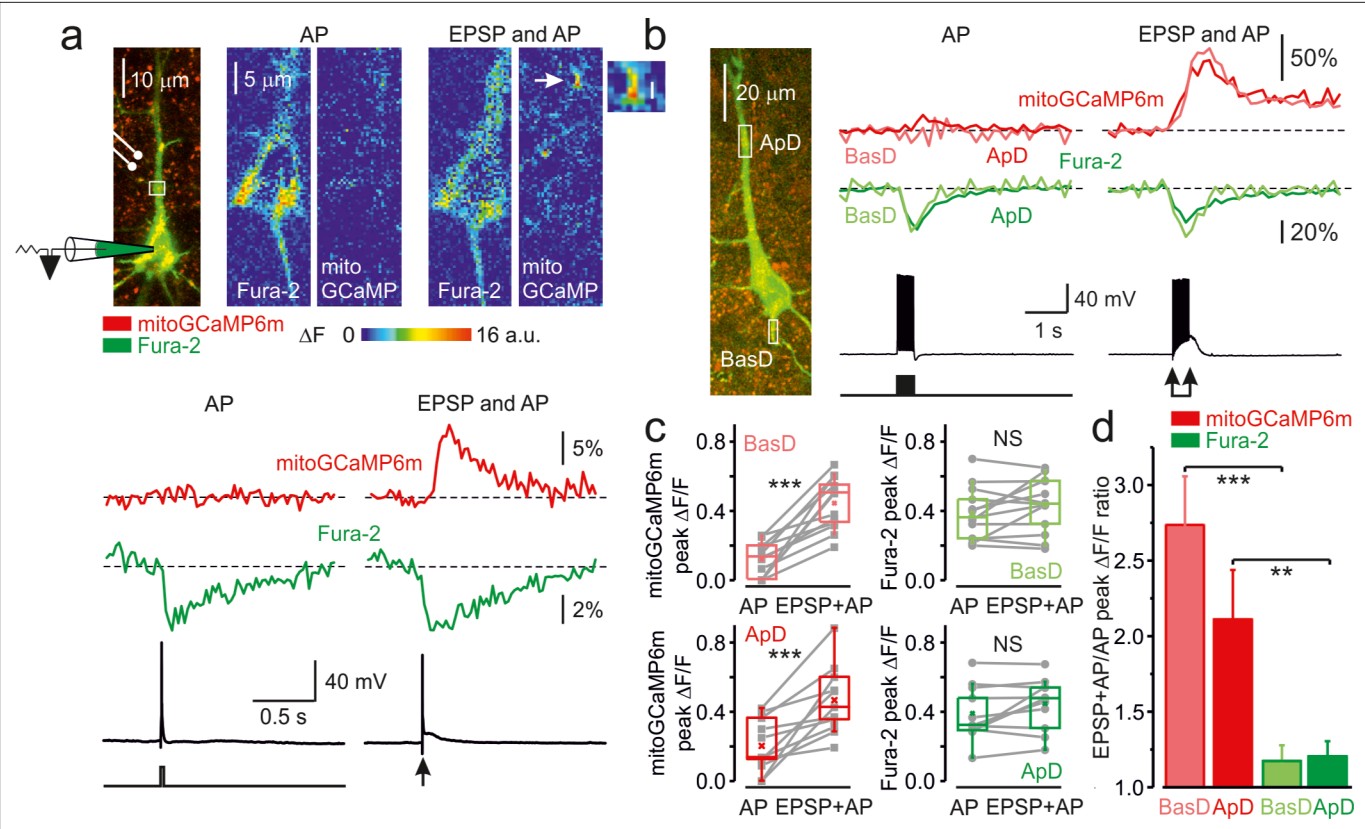

**Figure 4.** The coincidence of postsynaptic action potential and EPSP induces localized mitochondrial Ca²⁺ elevations in the dendrites. (**a**) Action potential elicited by a single, just suprathreshold synaptic stimulus induces a large, spatially restricted increase in dendritic mitoGCaMP6m fluorescence, whereas action potential evoked by a 5 ms current pulse (600 pA, cell body injection) had no such effect. *Top, Left*: Mitochondrial map in a representative L5 pyramidal neuron, obtained by merging the Fura-2 and mitoGCaMP6m fluorescence (for detail, see *Figure 1*). The rectangle indicates the region within the apical dendrite from which fluorescence measurements were obtained. *Right*: Pseudocolor maps of change in the mitoGCaMP6m and Fura-2 fluorescence in response to a synaptically and current pulse evoked AP. *Bottom*, The mitoGCaMP6m (red) and Fura-2 (green) ΔF/F transients measured from regions of interest as indicated in the upper panel and the somatic membrane potential trace. Arrow indicates a hotspot at which the synaptic stimulus elicited a mitochondrial Ca²⁺ transient. The transients are ensemble averages of 50 sweeps. (**b**) Action potentials elicited by a train of synaptic stimuli produce larger mitoGCaMP6m signals in the dendrites than APs elicited by the injection of a train of current pulses. *Left*: Mitochondria map in a representative L5 pyramidal neuron, obtained by merging the Fura-2 and mitoGCaMP6m fluorescence. The rectangles indicate the regions within the apical and basal dendrites from which fluorescence measurements were obtained. *Right*, Comparison of ΔF/F mitoGCaMP6m transients elicited in the apical (red) and basal dendrite (rose) by twenty suprathreshold synaptic stimuli at 50 Hz and by twenty brief current pulses delivered to the soma. The green and light green traces are ΔF/F Fura-2 transients in apical and basal dendrites, respectively. The black traces represent somatic membrane potential. (**c**) Peak amplitude of mitoGCaMP6m and Fura-2 ΔF/F transients elicited in basal (top) and apical dendrites (bottom) by a train of twenty suprathreshold current pulses or synaptic stimuli at a frequency of 50 Hz. The line connects the paired values obtained from the same individual neuron at two stimulation modalities. Box plots represent the 25–75% interquartile range, and the whiskers expand to the 5–95% range. A horizontal line inside the box represents the median of the distribution, and the mean is represented by a cross symbol (X). (**d**) Mean ratio of peak amplitudes of mitoGCaMP6m (rose, red) and Fura-2 (light green, green) ΔF/F transients elicited in dendrites by suprathreshold synaptic stimuli (EPSP + AP) and current pulses (AP). Data obtained from 8 to 13 individual neurons.

The online version of this article includes the following figure supplement(s) for figure 4:

**Figure supplement 1.** EPSPs alone cause no significant Ca²⁺ elevation in dendritic mitochondria.

**Figure supplement 2.** Availability of synaptic NMDA receptors is required for induction of dendritic mitochondrial Ca²⁺ elevation.

mitochondrial Ca²⁺ signals elicited by 20 unpaired APs and APs paired with EPSP. In both apical and basal dendrites, the paired APs produced a significantly larger mitochondrial signal. In contrast, the cytosolic Ca²⁺ elevation amplitude was not significantly different for unpaired and paired stimulation (*Figure 4b and c*). Hence, the peak amplitude of the mitoGCaMP6m transients elicited by paired APs was 2.73 ± 0.32 (n = 8) and 2.11 ± 0.32 (n = 9) times higher than of those evoked by the unpaired APs in the basal and apical dendrites, respectively (*Figure 4d*). The ability of postsynaptic neurons to

generate an AP was crucial for triggering the dendritic mito-Ca$^{2+}$ transients. Thus, intracellular dialysis with a solution containing a blocker of voltage-gated Na$^+$ channels, QX-314, which prevented the firing of the postsynaptic cells (*Connors and Prince, 1982*) while producing only a minor effect on the EPSP generation, dramatically reduced the amplitude of synaptically evoked mito-Ca$^{2+}$ transients (*Figure 4—figure supplement 1*). As with single subthreshold EPSPs, even the significantly larger EPSPs evoked in QX-314 dialyzed neurons by a train of strong synaptic stimuli, in the absence of AP, produced no significant Ca$^{2+}$ elevation in dendritic mitochondria.

It is well established that the main route of Ca$^{2+}$ entry into the spines is via the NMDARs (*Mainen et al., 1999*; *Miyazaki and Ross, 2017*), which require glutamate and depolarization to relieve the Mg$^{2+}$ block (*Nowak et al., 1984*). Blockade of NMDARs by bath applied APV (50 µM) almost completely and reversibly abolished the mitoGCaMP6m transients evoked by the paired APs in the dendrites (*Figure 4—figure supplement 2*).

## Discussion

Using simultaneous electrical recordings, cytosolic and mitochondrial Ca$^{2+}$ imaging in L5 pyramidal neurons, we found that relatively rare, singular spike firing, the firing mode associated with cortical circuit-based information processing in vivo (*Brecht and Sakmann, 2002*), produces fast-rising, rapidly decaying mitochondrial Ca$^{2+}$ transients in all neuronal compartments. In contrast to a recent report of 'loose' coupling of mitochondrial Ca$^{2+}$ transients to neuronal activity and cytosolic Ca$^{2+}$ transients in in vivo cortical neurons (*Lin et al., 2019*), our data suggest a tight, causative relationship between neuronal electrical activity, cytosolic and mitochondrial Ca$^{2+}$ levels.

Several studies (*Ashrafi et al., 2020*; *Devaraju et al., 2017*; *Kwon et al., 2016*; *Lewis et al., 2018*) have revealed the tens-of-seconds long mitochondrial Ca$^{2+}$ transients in presynaptic terminals of cultured central neurons. While this extremely slow rate of Ca$^{2+}$ clearance may reflect compartmental specialization of mitochondrial Ca$^{2+}$ handling at the presynaptic sites, it seems more likely to be due to excessively intense stimulation. Indeed, small mitochondrial Ca$^{2+}$ signals in axons of cultured hippocampal neurons elicited by 1–5 antidromic stimuli decayed rapidly (*Gazit et al., 2016*). Our results indicate that the prolonged mitochondrial Ca$^{2+}$ elevations occur only following the long period of robust high-frequency firing. Although such intense firing is not typically observed in cortical pyramidal cells under physiological conditions, it may occur during various neurological diseases (*Makinson et al., 2017*; *Zott et al., 2019*) contributing to the mitochondrial Ca$^{2+}$ overload and disease progression.

Our evidence indicates that the summation of the unitary AP-evoked mitochondrial Ca$^{2+}$ transients steeply depends on firing frequency. Thus, in contrast to cytosolic Ca$^{2+}$ signaling, which is less sensitive to firing rate, the mitochondrial Ca$^{2+}$ elevations were strongly amplified at firing frequencies of >40 Hz. As expected, the enhanced mitochondrial Ca$^{2+}$ uptake accelerated the rate of NAD(P)H production and consumption via the tricarboxylic acid cycle and the electron transport chain, respectively (*Díaz-García et al., 2021*), thereby upregulating the ATP synthesis.

The finding that action potential induced mitochondrial Ca$^{2+}$ transients are about threefold greater in the soma and apical dendrites than in proximal axon and basal dendrites, most likely, indicates the compartmental difference in expression of mitochondrial Ca$^{2+}$ channel, MCU. Alternatively, the subcellular differences in mitochondrial Ca$^{2+}$ handling may reflect region-specific MCU molecular tuning (*Ashrafi et al., 2020*; *Patron et al., 2019*) or mitochondrial morphology (*Lewis et al., 2018*). From the functional viewpoint, the greater mitochondrial Ca$^{2+}$ transients in thick neuronal processes might be necessary to compensate for the higher energetic cost of AP generation (*Attwell and Laughlin, 2001*).

A recent study suggests that the metabolic activity of mitochondria plays a pivotal role in synaptic plasticity (*Rangaraju et al., 2019*). How neuronal activity is linked to this process is poorly understood, however. Our results suggest that mitochondria can detect the Hebbian time coincidences between the pre- and postsynaptic spikes (*Markram et al., 1997b*). The resultant Ca$^{2+}$ elevations in the mitochondrial matrix could be an essential part of the cascade of events underlying spike-time-dependent synaptic plasticity. This cascade might involve the initiation of fission of the dendritic mitochondria, as has been proposed for chemically induced NMDAR-dependent LTP in hippocampal neuronal culture (*Divakaruni et al., 2018*), although it remains unclear whether the LTP associated burst of fission events occurs at a physiologically relevant time scale. The involvement of the frequency-dependent mitochondrial Ca$^{2+}$ signaling, most probably, explains the observed repetition rate requirement for LTP induction (*Inglebert et al., 2020*; *Lisman and Spruston, 2005*; *Sjöström et al., 2001*). Indeed,

the spike-timing-dependent potentiation of cortical synapses, in addition to precise spike timing, requires a sufficiently high repetition frequency (*Sjöström et al., 2001*). Our data indicate that the unique ability of the mitochondria to decode firing frequency and Hebbian timing code of neuronal activity make this organelle a long-thought link between the firing pattern, metabolism, and plasticity.

# Materials and methods

## Experimental animals

All experiments were approved by the Animal Care and Use Committee of Ben Gurion University of the Negev. C57BL/6 mice were obtained from Envigo (Israel).

## Viral constructs production and purification

cDNA of 2MT-GCaMP6m (mitoGCaMP6m) was subcloned by restriction/ligation (restriction enzymes and T4-ligase were from Fermentas/Thermo Scientific Life Science Research) into a plasmid containing adeno-associated virus 2 (AAV2) inverted terminal repeats flanking a cassette consisting of the neuronal-specific human synapsin one promoter (hSyn), the woodchuck post-transcriptional regulatory element (WPRE) and the bovine growth hormone polyA signal.

Viral particles were produced in HEK293T cells (ATCC) as previously described (*Tevet and Gitler, 2016*), using pAdDelta5 helper plasmids (a kind gift from Dr. Adi Mizrahi) and the pAAV2/9 n plasmid (Addgene #112865) which encode the rep/cap proteins of AAV2 and AAV9, respectively. Viral particles were then purified over iodixanol (Sigma-Aldrich) step gradients and concentrated using Amicon filters (EMD). Virus titers were measured by determining the number of DNase I–resistant vector genomes (vg) using qPCR with a linearized genome plasmid as a standard (*Challis et al., 2019*).

## Stereotaxic injections

Mice at the age of P21-25 were deeply anesthetized with Ketamine/Xylazine and then stereotactic bilateral injections were performed into the Layer 5 of the somatosensory cortex using a microliter syringe (Hamilton, Israel) at a rate of 0.25 µl/minute, with 500 nl of AAV9-hSyn-Mito-GCaMP6m containing $1 \times 10^{10}$ vg. After the injection, the needle was left in place for additional 3 min before being slowly removed from the brain. Coordinates for injections were (in mm): 4.1 rostral to lambda, ± 1.8 left/right of midline, –0.5 ventral to the pial surface.

## Acute coronal brain slices preparation

Coronal slices were prepared from mice three weeks post-injection (at the age of 6–7 weeks). The 300-µm-thick coronal cortical or horizontal hippocampal slices were prepared using standard techniques, as previously described (*Baranauskas et al., 2013*; *Fleidervish et al., 2010*). Mice were anesthetized with isoflurane (5%) and decapitated. The slices were cut on a vibratome (VT1200; Leica) and placed in a holding chamber containing oxygenated artificial cerebrospinal fluid (ACSF) at room temperature; they were transferred to a recording chamber after more than 1 hr of incubation. The composition of the ACSF (in mM): 124 NaCl, 3 KCl, 2 $CaCl_2$, 2 $MgSO_4$, 1.25 $NaH_2PO_4$, 26 $NaHCO_3$, and 10 glucose; pH 7.4 when saturated with 95% $O_2/CO_2$.

## Fluorescence imaging

Most experiments were performed on L5 pyramidal neurons in somatosensory neocortical slices. The cells were viewed with a 40 or 60× Olympus water-immersion lens of Ultima IV two-photon microscope (Bruker) equipped with a Mai Tai Deep See pulsed laser (Spectra-Physics). MitoGCaMP6m was excited at 940–950 nm. L5 pyramidal cells with low resting fluorescence that responded to electrical stimulation delivered via the nearby placed bipolar electrode were selected for whole cell recording (see below). In order to measure the cytosolic $Ca^{2+}$ transients, the intracellular solution was supplemented by $Ca^{2+}$ indicator, Fura-2 (100 µmol/l). The indicator was selected to minimize the interference with the mitoGCaMP6m fluorescence measurements. The Fura-2 fluorescence was elicited by two-photon excitation at 780 nm, and it declined as a function of the cytosolic $Ca^{2+}$ concentration. The neuronal morphology and the labelled mitochondria localization was obtained by scanning a Z-series of 30–40 high-resolution images at interval of 0.5 µm. The dynamic mitoGCaMP6m and Fura-2 imaging were performed from small regions of interest at frame rate of 10–50 Hz.

## Electrophysiology

Somatic whole-cell recordings were obtained using patch pipettes pulled from thick-walled borosilicate glass capillaries (1.5 mm outer diameter; Science Products, Germany). All recordings were at 30°C ± 0.5°C maintained with a temperature control unit (Luigs & Neumann, Rattingen). For current-clamp experiments the pipette solution contained (in mM): 130 K–gluconate, 6 KCl, 2 MgCl$_2$, 4 NaCl, and 10 Hepes, with pH adjusted to 7.25 with KOH. Pipettes had resistances of 5–7 MΩ when filled with this solution supplemented with Fura-2 (Molecular Probes). Recordings were made using a Multiclamp 700B amplifier (Molecular Devices) equipped with CV-7B headstage (Molecular Devices). Data were low–pass–filtered at 10 kHz (−3 dB), single-pole Bessel filter and digitized at 20 kHz using Digidata 1,322 A digitizer driven by PClamp 10 software (Molecular Devices). Care was taken to maintain the access resistance below 10 MΩ.

## Wide-field fluorescence imaging

The NAD(P)H auto-fluorescence signals were obtained using a 40× water-immersion lens (Olympus) in a BX51WI microscope (Olympus). The fluorescence was excited by using a high-intensity LED device (385 ± 4 nm, Prizmatix), and the emission was collected by using a modified Olympus U-MNU2 filter set (DC = 400 nm; EM = 420 nm). Images were collected with the Orca Flash 4.0 CMOS camera (Hamamatsu), using a pixel binning of 512 × 512, at a rate of 300ms per frame. A bipolar stimulating electrode (WPI, 0.01 MΩ) was placed ~100 µM below the region of interest, at the L-5/6 border in cortical slices or in CA1 stratum oriens in the hippocampal slices. The 0.1ms long extracellular stimuli were delivered using an optically coupled stimulus isolation unit (A.M.P.I) driven via the pClamp 10 software. Somatic whole-cell recordings (see above) were made from a pyramidal neuron in the middle of the region of interest. The stimulation intensity was carefully controlled so that each stimulus triggered only a single antidromic spike with a latency of <1ms post-stimulus. The baseline fluorescence was kept around 1500 a.u. throughout the experiments by regulating the intensity of LED emitted light.

## Data analysis

Electrophysiological data analysis was accomplished using pCLAMP10 software (Molecular Devices) and Origin 6.0 (OriginLab). The figures were created using CorelDraw X7 suite (Corel Corporation).

## Statistical analysis

If not otherwise noted, data are expressed as mean ± SE. Student *t*-test for paired or unpaired data was used for statistical analysis.

## Acknowledgements

This research was supported by The Israel Science Foundation (grant No. 1384/19 for IF, grant No.1763/21 for IS, and grant No. 1310/19 for DG) and by US National Institutes of Health grant (RF1 AG065628).

# Additional information

## Funding

| Funder | Grant reference number | Author |
| --- | --- | --- |
| Israel Science Foundation | 1384/19 | Ilya Fleidervish |
| Israel Science Foundation | 1310/19 | Daniel Gitler |
| National Institutes of Health | RF1 AG065628 | Israel Sekler Grace Stutzmann |
| Israel Science Foundation | 1763/21 | Israel Sekler |

The funders had no role in study design, data collection and interpretation, or the decision to submit the work for publication.

## Author contributions
Ohad Stoler, Data curation, Formal analysis, Investigation, Methodology, OS performed the electrical and imaging experiments, Validation, Visualization; Alexandra Stavsky, AS produced the viruses and performed stereotactic injections, Data curation, Formal analysis, Investigation, Validation, Visualization; Yana Khrapunsky, Data curation, Formal analysis, Investigation, Methodology, Project administration, Supervision; Israel Melamed, Formal analysis, Funding acquisition, Investigation, Writing - original draft; Grace Stutzmann, Formal analysis, Funding acquisition, Methodology, Supervision; Daniel Gitler, Formal analysis, Funding acquisition, Methodology, Resources, Supervision; Israel Sekler, Conceptualization, Data curation, Formal analysis, Funding acquisition, Investigation, Methodology, Resources, Supervision, Validation, Visualization, Writing - original draft, Writing - review and editing; Ilya Fleidervish, Conceptualization, Data curation, Formal analysis, Funding acquisition, Investigation, Methodology, Resources, Supervision, Validation, Writing - original draft, Writing - review and editing

## Author ORCIDs
Ohad Stoler http://orcid.org/0000-0002-3420-8382
Alexandra Stavsky http://orcid.org/0000-0002-8209-3524
Daniel Gitler http://orcid.org/0000-0001-9544-3610
Ilya Fleidervish http://orcid.org/0000-0002-5501-726X

## Ethics
All experiments were approved by the Animal Care and Use Committee of Ben Gurion University of the Negev (protocol #IL-68-09-2019).

## Decision letter and Author response
Decision letter https://doi.org/10.7554/eLife.74606.sa1
Author response https://doi.org/10.7554/eLife.74606.sa2

---

# Additional files

## Supplementary files
• Transparent reporting form

## Data availability
A representative subset of the raw electrical recording and imaging data has been deposited to Dryad (https://doi.org/10.5061/dryad.sxksn0348). The dataset contains the Microcal Origin opj files of the electrical and optical recordings and quantitative analysis of the data. We are unable to make all raw electrical recording and imaging data publicly available as due to the large size of our raw dataset (>10TB). Interested researchers should contact the corresponding author to gain access to the raw data.

The following dataset was generated:

| Author(s) | Year | Dataset title | Dataset URL | Database and Identifier |
|---|---|---|---|---|
| Fleidervish I, Stoler O, Stavsky A, Khrapunsky Y, Melamed I, Stutzmann G, Gitler D, Sekler I | 2022 | Frequency- and spike-timing-dependent mitochondrial Ca2+ signaling regulates the metabolic rate and synaptic efficacy in cortical neurons | https://dx.doi.org/10.5061/dryad.sxksn0348 | Dryad Digital Repository, 10.5061/dryad.sxksn0348 |

---

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
