## [Decision Letter]

**Decision letter after peer review:**

Thank you for submitting your article "Frequency-and spike-timing-dependent mitochondrial Ca^2+^ signaling regulates the metabolic rate and synaptic efficacy in cortical neurons" for consideration by *eLife*. Your article has been reviewed by 3 peer reviewers, including Sacha B Nelson as the Reviewing Editor and Reviewer #1, and the evaluation has been overseen by John Huguenard as the Senior Editor. The following individual involved in review of your submission has agreed to reveal their identity: Bruce P. Bean (Reviewer #2).

Essential revisions:

1) Please modify the abstract and discussion to ensure that readers are not given the impression that the role of the observed mitochondrial calcium signals in long term synaptic plasticity have been directly tested in this manuscript

2) Please enhance the discussion along the lines suggested by Reviewer 3.

*Reviewer #3 (Recommendations for the authors):*

The main current weaknesses of the study are the lack of mechanistic insight and the mainly observational data. The authors don't show that calcium uptake into mitochondria drives the frequency dependent metabolic change. Similarly their proposal that mitochondrial co-incidence detection could impact plasticity is not tested experimentally. Addressing some of these issues would significantly increase the impact of the findings.

The authors propose that in different neuronal subdomains, mitochondrial calcium uptake may respond differently to activity. While this is interesting, no mechanism is provided for this.

The authors report a frequency dependency of mitochondrial calcium uptake. Given that calcium uptake through MCU may be significantly impacted by calcium concentration at the mitochondrial OMM, this seems rather an expected result. That is, increased firing rates will lead to faster and higher calcium rises that could influence the cooperativity of MCU uptake mechanisms. The authors show that cytoplasmic calcium also significantly goes up (e.g. between 20 and 50 APs; Figure 2B) so isn't the facilitated mitochondrial calcium uptake at higher frequencies simply a consequence of this?

The authors nicely demonstrate that there is also a frequency dependency of metabolic rate (imaging NAD(P)H) but not evidence is provided that this is indeed dependent on mitochondrial calcium influx. Other mechanisms could be at play. Can the authors provide evidence (pharmacological, genetic) that frequency-dependent metabolic rate change is indeed dependent on mitochondrial calcium influx.

The examples shown for mitochondrial co-incidence detection appear to be in the proximal dendrite 20-40 microns from the cell body. Backpropagating APs can travel many hundreds of microns into dendrites. Have the authors observed similar phenomena more distally?

The discussion is overly succinct and therefore fails to place the current work within the wider literature. A number of studies in axons already not only studied activity-dependent mitochondrial calcium uptake in detail but also (unlike the current work) looked at its physiological role (e.g. Kwon et al., 2016; Gazit et al., 2016; Vaccaro et al.., 2017; Lewis et al., 2018; Devaraju et al., 2017; Styr et al., 2019; Ashrafi et al., 2020 to mention just some of the most recent ones). Notably, Divakaruni et al., 2018 previously demonstrated (chemical) LTP induced mitochondrial calcium transients but this work is not discussed despite the authors of the current study proposing the importance of mitochondrial calcium uptake for Hebbian plasticity.

---

## [Author Response]

Essential revisions:1) Please modify the abstract and discussion to ensure that readers are not given the impression that the role of the observed mitochondrial calcium signals in long term synaptic plasticity have been directly tested in this manuscript

Thank you for raising this important issue. The abstract and discussion are now modified accordingly.

2) Please enhance the discussion along the lines suggested by Reviewer 3.

The Discussion is extended along the lines suggested by Reviewer 3. We added discussion of the earlier studies of activity-dependent mitochondrial calcium uptake, as recommended by the Reviewer (Kwon et al., 2016; Gazit et al., 2016; Vaccaro et al., 2017; Lewis et al., 2018; Devaraju et al., 2017; Styr et al., 2019; Ashrafi et al., 2020). We also added discussion of the results by Divakaruni et al. (2018), who proposed a role of fission of the dendritic mitochondria as a mechanism of the chemically induced NMDAR-dependent LTP in hippocampal neuronal culture.

Reviewer #3 (Recommendations for the authors):The main current weaknesses of the study are the lack of mechanistic insight and the mainly observational data. The authors don't show that calcium uptake into mitochondria drives the frequency dependent metabolic change. Similarly their proposal that mitochondrial co-incidence detection could impact plasticity is not tested experimentally. Addressing some of these issues would significantly increase the impact of the findings.

We agree with the Reviewer that elucidating the molecular mechanisms regulating the Ca^2+^ homeostasis of neuronal mitochondrial is of utmost importance. We plan to address these mechanisms in future work. We anticipate that the publication of our data will stimulate other labs in the field to contribute effort to advance our understanding of the activity-metabolic relationship in central neurons.

The authors propose that in different neuronal subdomains, mitochondrial calcium uptake may respond differently to activity. While this is interesting, no mechanism is provided for this.

We hypothesize that the observed subcellular differences in the magnitude of mitochondrial Ca^2+^ transients are related to compartmentally specific regulation of MCU expression or its activity (see Discussion). The detailed analysis would require developing new, currently unavailable molecular tools suitable for in-vivo manipulations. We plan to address this important issue in future studies.

The authors report a frequency dependency of mitochondrial calcium uptake. Given that calcium uptake through MCU may be significantly impacted by calcium concentration at the mitochondrial OMM, this seems rather an expected result. That is, increased firing rates will lead to faster and higher calcium rises that could influence the cooperativity of MCU uptake mechanisms. The authors show that cytoplasmic calcium also significantly goes up (e.g. between 20 and 50 APs; Figure 2B) so isn't the facilitated mitochondrial calcium uptake at higher frequencies simply a consequence of this?

We respectfully disagree. To the best of our knowledge, the spike-frequency-dependent amplification of mitochondrial Ca^2+^ signaling has never been demonstrated in neurons or other cell types. We believe that a dramatic increase in the amplitude of mitochondrial Ca^2+^ response at 50 vs. 20 Hz cannot be explained by slightly more prominent (10-20%, Figure 2B, Figure 2—figure supplement 1) cytosolic Ca^2+^ elevation. The Ca^2+^ microdomains beneath the OMM, 5- to 10-fold higher than in bulk cytosol, have indeed been demonstrated by the Pozzan’s group (Giacomello et al., 2010). However, there is no evidence that the Ca^2+^ concentration at these hotspots behaves in a non-linear fashion relative to the bulk.

The authors nicely demonstrate that there is also a frequency dependency of metabolic rate (imaging NAD(P)H) but not evidence is provided that this is indeed dependent on mitochondrial calcium influx. Other mechanisms could be at play. Can the authors provide evidence (pharmacological, genetic) that frequency-dependent metabolic rate change is indeed dependent on mitochondrial calcium influx.

Thank you for raising this important issue. Several other groups have shown the activity-dependent mitochondrial Ca^2+^ elevation to promote the oxidation and recovery of mitochondrial NAD(P)H. Most recently, an elegant *eLife* paper (Diaz-Garcia, 2021 #64) showed a clear link between mitochondrial Ca^2+^ signaling and NAD(P)H changes. These experiments were performed under similar experimental conditions, and therefore we felt that it would be redundant to repeat them here. Indeed, our results are entirely consistent with this study. From a historical perspective, the close correlation of mitochondrial Ca^2+^ signaling and NAD(P)H changes was demonstrated as early as in 1992 when Duchen (Duchen, 1992) reported that the superfusion with Ca^2+^-free solution or intracellular application of the MCU blocker, RuR, completely abolished changes in NADH autofluorescence.

The examples shown for mitochondrial co-incidence detection appear to be in the proximal dendrite 20-40 microns from the cell body. Backpropagating APs can travel many hundreds of microns into dendrites. Have the authors observed similar phenomena more distally?

Our conclusions are based on recordings from the mitochondria in proximal apical and basal dendrites, up to 200 µm from the soma. The dynamics of backpropagation and Ca^2+^ signaling could indeed differ in distal vs. proximal dendritic arbor. We believe, however, that our finding that the coincidence of pre- and postsynaptic spikes cause spatially restricted mitochondrial Ca^2+^ signals will remain valid for distant dendrites.

The discussion is overly succinct and therefore fails to place the current work within the wider literature. A number of studies in axons already not only studied activity-dependent mitochondrial calcium uptake in detail but also (unlike the current work) looked at its physiological role (e.g. Kwon et al., 2016; Gazit et al., 2016; Vaccaro et al.., 2017; Lewis et al., 2018; Devaraju et al., 2017; Styr et al., 2019; Ashrafi et al., 2020 to mention just some of the most recent ones). Notably, Divakaruni et al., 2018 previously demonstrated (chemical) LTP induced mitochondrial calcium transients but this work is not discussed despite the authors of the current study proposing the importance of mitochondrial calcium uptake for Hebbian plasticity.

Thank you for raising this important issue. We have now extended the Discussion, relating our results to those mentioned above.

References:

Duchen, M.R. (1992). Ca(2+)-dependent changes in the mitochondrial energetics in single dissociated mouse sensory neurons. Biochem J 283 ( Pt 1), 41-50.

Giacomello, M., Drago, I., Bortolozzi, M., Scorzeto, M., Gianelle, A., Pizzo, P., and Pozzan, T. (2010). Ca^2+^ hot spots on the mitochondrial surface are generated by Ca^2+^ mobilization from stores, but not by activation of store-operated Ca^2+^ channels. Mol Cell 38, 280-290.

Miyazaki, K., and Ross, W.N. (2017). Sodium Dynamics in Pyramidal Neuron Dendritic Spines: Synaptically Evoked Entry Predominantly through AMPA Receptors and Removal by Diffusion. J Neurosci 37, 9964-9976.